# Towards Robust Delivery of Antimicrobial Peptides to Combat Bacterial Resistance

**DOI:** 10.3390/molecules25133048

**Published:** 2020-07-03

**Authors:** Matthew Drayton, Jayachandran N. Kizhakkedathu, Suzana K. Straus

**Affiliations:** 1Department of Chemistry, University of British Columbia, 2036 Main Mall, Vancouver, BC V6T 1Z1, Canada; mdrayton@chem.ubc.ca; 2Department of Pathology and Laboratory Medicine, and Centre for Blood Research, University of British Columbia, 2350 Health Sciences Mall, Life Sciences Centre, Vancouver, BC V6T 1Z3, Canada; jay@pathology.ubc.ca

**Keywords:** antimicrobial peptide (AMP), multidrug resistance (MDR), delivery vehicles, bioconjugation, encapsulation

## Abstract

Antimicrobial peptides (AMPs), otherwise known as host defence peptides (HDPs), are naturally occurring biomolecules expressed by a large array of species across the phylogenetic kingdoms. They have great potential to combat microbial infections by directly killing or inhibiting bacterial activity and/or by modulating the immune response of the host. Due to their multimodal properties, broad spectrum activity, and minimal resistance generation, these peptides have emerged as a promising response to the rapidly concerning problem of multidrug resistance (MDR). However, their therapeutic efficacy is limited by a number of factors, including rapid degradation, systemic toxicity, and low bioavailability. As such, many strategies have been developed to mitigate these limitations, such as peptide modification and delivery vehicle conjugation/encapsulation. Oftentimes, however, particularly in the case of the latter, this can hinder the activity of the parent AMP. Here, we review current delivery strategies used for AMP formulation, focusing on methodologies utilized for targeted infection site release of AMPs. This specificity unites the improved biocompatibility of the delivery vehicle with the unhindered activity of the free AMP, providing a promising means to effectively translate AMP therapy into clinical practice.

## 1. Introduction

Antimicrobial peptides (AMPs), also known as host defence peptides (HDPs), have garnered a recent surge in interest as weapons in the fight against multidrug resistant (MDR) bacteria due to their broad spectrum activity, multimodal functionalities, and minimal resistance generation. Indeed, many reviews regarding AMPs and their potential as drug candidates have been published in the last couple of years [1,2,3,4]. These compounds are polypeptide sequences (typically 12–50 residues in length) consisting of cationic and hydrophobic amino acids. They are considered to be interesting substitutes to antibiotics because they can function in a variety of ways [2,3,5]. They act against pathogenic species, such as viruses, fungi, and parasites [5,6,7] and can display anti-cancer activity [8,9,10] or modulate the immune response [11,12,13]. HDPs can directly kill Gram-positive and/or Gram-negative bacteria in the planktonic (i.e., free-swimming) or biofilm form [4,14,15]. Increasingly, HDPs are being found to possess more than one of these functions at a time [2,8,16]. Notably, Mookherjee et al. [1] detailed the mechanisms of action of AMPs against infection both by direct bacterial destruction and by modulation of the host’s immune response, and touched upon the clinical potential of these AMP-based therapies.

Despite all their promise, AMPs have not found wide spread use in the clinic for a number of reasons. Firstly, because most AMPs consist of l-amino acids, they are susceptible to protease degradation and rapid kidney clearance [17,18,19]. Furthermore, some AMPs are not specific to bacteria and hence display systemic toxicity. Indeed, oral administration of AMPs can lead to proteolytic digestion by enzymes in the digestive tract such as trypsin and pepsin. Moreover, systemic administration results in short half-lives in vivo and cytotoxic profiles in blood [11]. Many strategies have been investigated to circumvent these issues [3]. These include the chemical modification of AMPs [20] and the use of delivery vehicles [21]. Recently, Martin-Serrano et al. [22] and Makowski et al. [23] highlighted the nanoparticle and lipid systems being developed for AMP formulation. Here, we provide a broader view of the methodologies being utilized to improve the properties of AMPs for their translation into clinical use, highlighting recent developments in delivery vehicle formulation as well as peptide modification, while focusing on the utilization of targeting and release mechanisms for site-specific treatment of infection (Figure 1). The infection site release of AMPs marries the improved biocompatibility of the delivery vehicle with the unhindered activity of the free peptide, thereby reducing off-site toxicity and rapid degradation of the free peptide while maintaining high peptide concentration at the location of interest.

This review does not focus on specific AMP classes or subtypes, but rather examines a range of delivery vehicles applied to a broad definition of AMPs (i.e., linear peptides, lipopeptides, glycopeptides). The review also presents alternatives to delivery vehicles (i.e., d-amino acid substitution, lipidation, rational amino acid substitution, cyclization/stapling and peptidomimetics). Further, as the majority of AMPs being tested clinically are for topical administration where concerns regarding degradation, toxicity and specificity are lessened [24,25], we focus mainly on systemically administered therapies that would benefit most from these advanced delivery systems, though some topical systems are also mentioned. Indeed, AMPs are considered viable alternatives to conventional antibiotics and have gathered significant support for clinical evaluation, but the difficulties associated with systemic therapy necessitate additional formulation designs to improve stability and efficacy and to decrease toxicity [25]. By examining the various strategies that are already available, this review aims to inspire the future development of more robust delivery strategies.

## 2. Targeting of Delivery Vehicles

When utilizing delivery systems for drug delivery, oftentimes the payload is directly conjugated to or encapsulated in the delivery vehicle without a release mechanism. Indeed, the majority of delivery systems discussed in this review utilize this blueprint. Though this delivery system can improve the drug’s biocompatibility, stability and pharmacokinetics greatly, it often hampers its efficacy. As a partial solution, many formulations utilize non-specific release (e.g., hydrolytic release) for prolonged drug delivery, but this can raise concerns regarding off-site toxicity, particularly as high drug dosages are required to maintain suitable concentration at the target site [26]. Thus, there has been great interest recently in developing strategies to release the payload from the delivery vehicle at the targeted disease site—in the case for AMPs discussed herein, the site of infection. In this regard, infection-site release can be achieved by targeting the unique biochemical microenvironment associated with infected tissue (i.e., endogenous targeting), or by extrinsic guidance and spatiotemporal release by external stimuli (i.e., exogenous targeting). Below, we discuss both endogenous and exogenous targeting, as well as passive targeting, highlighting examples used for AMP delivery (Figure 1). Though few examples of targeted AMP release have been published in literature to date, such release mechanisms can be utilized to improve the specificity, stability, and activity of AMP therapeutics, as discussed below.

### 2.1. Passive Targeting

Hydrolytic cleavage is frequently used to allow for the continuous, prolonged release of payloads that can range from days to months [26]. This is generally accomplished by incorporating ester [26] or ketal [27] linkages between the delivery vehicle and the drug. However, this raises concerns regarding off-site toxicity and dosage requirements. Alternatively, a process known as passive targeting can be utilized for nanodelivery systems due to the increased vascular permeability associated with bacterial infection [28,29,30]. This permeability arises from the local release of agents from dead or live bacteria, such as lipopolysaccharide and lipoteichoic acid, which trigger inflammation and activate immune cells, resulting in loose endothelial junctions and, correspondingly, leaky vasculature. Furthermore, the resulting impaired lymphatic drainage associated with infection can potentially enable passive accumulation of the drug at the infection site [28]. This is known as the enhanced permeability and retention (EPR) effect, and has been studied extensively for the treatment of cancer, which also shows inflammatory conditions similar to bacterial infection [28,29]. A number of nanoparticles have been found to accumulate in infected tissue in this way; for example, polyethylene glycol (PEG)-coated liposomes have exhibited this phenomenon in mouse soft tissue infected by *Staphylococcus aureus* [31,32,33].

### 2.2. Endogenous Targeting

For direct endogenous targeting, specific infection-associated ligands can be exploited. Vancomycin [34,35], antibodies targeting bacterial surface proteins [36], aptamers [37,38], and lectins [39,40] have all been conjugated to nanoparticles for bacteria-specific targeting. For example, conjugating an antibody specifically binding to staphylococcal protein A, a species-specific surface protein, to daptomycin-loaded polydopamine-coated gold nanocages enabled targeting of *S. aureus* with no binding to mammalian cells [36]. However, ligand-based targeting can be limited by the accessibility of the target cells as well the structural heterogeneity of the targeted molecules [41]. Similarly, the negative surface charge of bacteria can provide a means of targeting through the electrostatic interactions of cationic nanoparticles [28]. In this is one way, AMPs themselves are being designed to promote specificity.

A number of characteristics unique to the microenvironment of the infection site can also be utilized for targeted delivery of antibiotics. These include pH, redox gradients, and enzyme concentration. Firstly, pH-sensitive linkers can take advantage of the local low pH environment associated with bacterial infection and biofilms. This pH can reach as low as 4.5 in the case of biofilms [42,43,44] and is associated with anaerobic fermentation and inflammation, both of which produce acidic products [41,42]. Targeting infection-associated pH allows for indiscriminate bacterial targeting, which can be useful in broad-spectrum targeting. For example, Radovic-Morena et al. developed vancomycin-encapsulating pH-responsive poly(d,l-lactic-*co*-glycolic acid)-*b*-poly(L-histidine-*b*-poly(ethylene glycol) nanoparticles enabling pH-sensitive binding to bacteria [42]. A surface charge switch resulting from the protonation of histidine imidazole groups at low pH resulted in strong electrostatic interactions with the negatively charged bacterial cell surfaces. This method of delivery additionally mitigated the decrease in vancomycin activity associated with lower pH, potentially due to the interactions of the nanoparticles themselves with the bacteria. The resulting minimum inhibitory concentrations (MICs) of free vancomycin and the pH-responsive vancomycin nanoparticles were 2.4 and 8.6 μg/mL at pH 6.0, denoting 2.0- and 1.3-fold increases from pH 7.4, respectively. Chu et al. utilized a similar strategy for vancomycin release using pH-sensitive PEG-polycaprolactone-poly(β-amino ester) triblock polymers [45].

Chemical moieties such as hydrazone, aconityl and acetal/ketal linkages are stable in blood circulation but undergo rapid hydrolysis at lower pH, potentially allowing payload release at infection sites [26,27,46,47,48,49]. These are used heavily for the selective release of drugs in mildly acidic tumour environments [26]. Furthermore, these linkers can be used for tissue-specific release in the case of the gastrointestinal tract, vagina and skin [42], where bacteria often inhabit, and for intracellular release (vide infra). However, these linkers seem to exhibit low serum stability compared to other linker types. To the best of our knowledge, no AMP bioconjugates utilizing this method for infection site release have thus far been reported. Alternatively, redox-sensitive release can potentially be harnessed for targeting infection due to the accumulation of reactive oxygen species (ROS) in inflammatory tissue [41,50].

Finally, enzymatically cleavable linkers enable targeting of both enzymes secreted by the bacteria themselves and/or enzymes released by the host in response to infection. The altered expression of these enzymes (e.g., proteases, lipases and glycosidases) can allow for drug accumulation at the site of infection [41]. Protease targeting is particularly appealing for AMP development as the sequences can be directly adjoined to the *N-* or *C*-terminus of the peptide during expression or chemical synthesis. For example, gelatinase, an enzyme secreted by a number of bacteria, including *S. aureus* [51,52], and bacterial lipases [53] have been examined for this purpose. Enzymes secreted by the host during the inflammatory response to infection can also potentially be harnessed here (e.g., matrix metalloproteinases [54]).

#### Targeting Intracellular Bacteria

It is becoming increasingly apparent that intracellular bacteria play a large role in recurrent and prolonged bacterial infection. During infection, the body’s most immediate defence against pathogenic bacteria is the innate immune system [55]. This system houses a number of immune cells that recognize the invading pathogen and remove it from the body. In particular, macrophages and neutrophils, phagocytic cells acting as the first line of defence against infection, engulf bacteria within minutes of infection [56]. The engulfed bacteria are then destroyed by fusion with acidic lysosomes, which deliver digestive enzymes, bactericidal proteins (e.g., lysozyme), proton pumps, ROS and reactive nitrogen species to the phagosome [57,58].

Although phagolysosomal killing is usually quite effective, a number of bacteria, including *Escherichia coli* and *S. aureus*, have developed methods to survive this process, allowing them to reside within the cell for prolonged periods of time [56,58]. These pathogen-containing immune cells can then disseminate the bacteria throughout the body, where they can infect other tissues, thereby causing chronic or recurrent infections [56,59]. Treatment of intracellular bacteria can be particularly troublesome due to their location within the cell, as they can avoid antibiotics unable to cross the cell membrane and can enter dormant states (e.g., small colony variants and persister cells) that alter their susceptibility to antibiotics [60].

However, the unique biochemistry of the intracellular environment can be harnessed for treatment of these bacteria by intracellular payload release. Linker moieties such as disulphides and thioethers can allow for cleavage within the highly reducing environment of the cell, while the acid-sensitive linkers detailed earlier can allow for release after uptake by acidic endosomes and lysosomes [26,41]. Furthermore, linkers susceptible to cleavage by phagolysosomal proteases, such as cathepsin B, can be utilized. Though no AMP conjugates have been utilized to target intracellular bacteria thus far to our knowledge, certain AMPs (e.g., LL-37) have been shown to be effective against intracellular bacterial strains including *S. aureus* [61] and Mycobacteria [62,63], suggesting they could be promising therapeutics for intracellular bacteria killing.

One unique approach to targeting bacterial infection is to take advantage of the innate targeting of phagocytic immune cells that spontaneously scavenge and destroy bacteria within the body [28,58,64]. For example, Xiong et al. developed vancomycin-loaded nanogels covered in mannosyl ligands for targeted delivery to macrophages, which express high levels of mannose receptors [64]. Once the nanogel-containing macrophages engulfed bacteria at the site of infection, the nanogel’s polyphosphoester core was degraded by bacterially produced phosphatase or phospholipase, causing release of the vancomycin and subsequent bacterial destruction. More recently, in a method using adoptive macrophage therapy, Hou et al. utilized vitamin lipid nanoparticles to deliver an mRNA strand encoding an AMP-cathepsin B conjugate to cultured macrophages [58]. When delivered to the cytosol of the cell, the mRNA was translated, and the resulting protein conjugate was subsequently trafficked to the lysosome by its cathepsin B tag, where it was cleaved by the enzyme to release the AMP. These macrophages, enhanced with a lysosomally localized broad spectrum AMP, displayed efficient killing of drug resistant intracellular *E. coli* and *S. aureus* and recovered the immune system of immunocompromised septic mice.

### 2.3. Exogenous Targeting

Externally applied (exogenous) stimuli can also be utilized for drug delivery. Unlike endogenous targets, these responsive carriers do not have an innate localization method, so localization to the site of infection is often accomplished by targeting ligands or by delivering the vehicles to the site by external means, such as magnetic guidance. Once there, the payload is released by an externally applied stimulus, such as temperature changes, electric or magnetic fields, ultrasound, light, lasers, etc. [41,65]. For example, Meeker et al. developed photoactivatable daptomycin-loaded polydopamine-coated gold nanocages conjugated with *S. aureus*-targeting antibodies [36]. These nanocages released daptomycin in response to near-infrared light irradiation, which itself generated localized bactericidal photothermal effects, thereby combining the localized release of antibiotic with photothermal therapy to eradicate both planktonic and biofilm-associated *S. aureus.* The authors suggested further development could allow for potential applications in the treatment of orthopedic infections (e.g., surgical debridement), where surgeons would have direct access to the site to facilitate laser irradiation. Similarly, thermo-responsive polymers have been explored for the controlled delivery of alamethicin [66], as have nanoparticle-embedded chitosan microbeads for the magnetic release of vancomycin [67]. Maleki et al. also developed antimicrobially active gold-coated superparamagnetic iron oxide nanoparticles conjugated to the hybrid AMP cecropin mellitin for the potential translation to site-specific targeting via external magnetic stimuli [68].

## 3. Delivery Vehicles

Delivery vehicles are often harnessed for drug delivery to improve the biological properties of the therapeutic. These vehicles generally aim to improve the biocompatibility, stability, solubility, circulation time, and pharmacokinetics of the drug. They are particularly useful for AMPs, which often innately exhibit high cytotoxicity, unpredictable activity due to protease susceptibility, and short circulation time as a result of rapid degradation by blood proteases and removal by kidney filtration or the reticuloendothelial system (RES) [69], which is highlighted by the limited number of systemically administered AMPs in clinical trials [24,25]. Herein, we discuss the delivery vehicles developed in recent years for the encapsulation or conjugation of AMPs for combatting bacterial infection. A number of advantages and disadvantages of the delivery vehicles presented herein are summarized in Table 1.

### 3.1. Lipid Encapsulations

Unlike other delivery vehicles that require covalent conjugation of AMPs, potentially the most common vehicle for AMP delivery is lipid encapsulation. These formulations, ranging from liposomes and micelles to liquid crystalline nanoparticles, are termed lipid nanoparticles (LNPs) as a result of their size. Due to the natural lipid molecules utilized in their formulation, LNPs generally benefit from high biocompatibility, safety, and biodegradability, which alleviates concerns regarding cytotoxicity and organ accumulation that potentially arise with other nanoparticles [22,23]. In addition, their amphiphilic nature allows them to encapsulate both hydrophilic and hydrophobic molecules, making them particularly valuable for AMPs [22]. Interestingly, certain lipid nanostructures, such as liposomes, reconstituted lipoproteins, and cell membrane-derived nanostructures, have also recently gained attention as “nanodecoys” to trap and restrain pathogens and their toxins to treat infectious diseases [86], potentially providing another advantage for AMP formulation. An example of these agents is CAL02, a clinically tested liposomal antivirulence drug that entraps and neutralizes a large array of virulence factors produced by both Gram-positive and Gram-negative bacteria [87]. By mimicking the cell membrane lipid composition that many virulence effectors target, these liposomes can bind to these toxins with higher affinity than cells, thereby preventing the threatening complications these molecules are associated with during severe infections (e.g., tissue damage and organ failure).

#### 3.1.1. Liposomes

Used for decades in drug delivery, liposomes are lipid bilayer vesicles that contain a hollow, aqueous center. The bilayer is generally comprised of phospholipids, though other lipids and membrane components, such as cholesterol, are often added to improve their physiochemical properties. Due to their amphiphilic nature, the phospholipids spontaneously form in aqueous solution, with their polar head groups facing the outside/inside of the bilayer and their nonpolar fatty acid tails comprising the interior of the bilayer [23] (Figure 1).

The size, lipid composition, and surface modifications of liposomes can be tuned extensively to alter their physiochemical properties [22]. For example, lipid composition greatly influences the packing, fluidity, and charge of the liposome, which can affect both stability and encapsulation efficiency. In particular, size plays an important role in the efficacy of liposomes. Though they range from 50 to 500 nm in diameter, liposomes smaller than 200 nm are potentially best used for infection and other inflammatory diseases, as they have been shown to accumulate at the target site due to the previously described EPR effect [23,31,32,33]. In addition, liposomes can be coated with other materials to improve their properties [88,89,90,91]. For example, coating AMP-containing liposomes with a nontoxic, antimicrobial cationic polymer increased the stability of the liposomes.

While also significantly improving the activity of the encapsulated peptide up to 2000-fold against food-borne pathogenic bacteria [89]. In addition, coating liposomes with neutral polymers such as PEG has been shown to greatly increase their stability and circulation time [92,93].

For clinical practice, liposomes suffer from two major drawbacks: i. low in vivo stability due to phagocytic cell clearance, and ii. limited practical sterilization techniques for scaling-up [22,94]. Though improvements in the latter are being made with advancements in various technologies, particularly microfluidics [95,96,97], other delivery vehicles have become increasingly more popular for AMP delivery.

#### 3.1.2. Micelles

Similar to liposomes, micelles are another promising lipid-based carrier for AMP delivery. The amphiphilic nature of a micelle allows for high loading capacity of low water-soluble therapeutics in its hydrophobic inner core and improved in vivo circulation by reducing RES uptake via its hydrophilic outer corona [98]. Furthermore, micelles tend to display higher stabilities than liposomes and respond more strongly to stimuli than nanoparticles [72]. Most commonly, PEGylated phospholipids, such as 1,2-distearoyl-sn-glycero-3-phosphoethanolamine- *N*-[amino(polyethylene glycol)-2000] (DSPE-PEG2000), are utilized for this purpose, as they spontaneously form stabilized micelles with a hydrophobic DSPE core surrounded by hydrophilic PEG molecules [99]. Recently, our groups utilized DSPE-PEG2000 micelles as a delivery vehicle for AMPs derived from aurein 2.2 [99]. The peptide, along with its d-amino acid, retro-inverso and *C*-terminus cysteine-containing versions, showed marked improvement in their hemolytic activity and cytotoxicity when encapsulated in DSPE-PEG2000 micelles. Furthermore, the peptide formulations displayed no aggregation at concentrations as high as 7.5 mg/kg in an in vivo mouse abscess infection model, where they were administered by subcutaneous injection. Though all encapsulated peptides showed low to moderate decreases in activity in vitro, they all maintained antimicrobial activity at 5 mg/kg in vivo, with the most promising peptide reducing abscess size by 85% and bacterial count 510-fold. These results highlight the potential of micelles for AMP delivery and also reveal the discrepancies between in vitro and in vivo activity that can result in potentially clinically efficacious AMPs not being tested in vivo. Similar work done by Lee et al. found that DSPE-PEG2000 micelles conjugated on their surface to a cationic antimicrobial peptide improved the survival rate of septic mice [100].

#### 3.1.3. Liquid Crystalline Nanoparticles

Recently, lyotropic liquid crystalline nanoparticles (LCNPs) have been explored for AMP formulation. These particles are similar to liposomes in that they are comprised of lipid bilayers, but they are structured into highly organized crystal-like two- and three-dimensional nonlamellar nanostructures that confer high thermal stability [22,23,101]. They are often termed cubosomes or hexasomes depending on the dimensions of their internal phases [101]. Boge et al. found that encapsulation of LL-37 in cubosomes substantially protected the AMP from proteolytic degradation but was accompanied by a loss in the AMP’s broad-spectrum activity, with only activity against Gram-negative bacteria remaining after encapsulation [102]. In 2019, these researchers further optimized the use of cubosomes for topical LL-37 delivery by utilizing three different preparation protocols [103]. Here, they confirmed the encapsulated LL-37 retained activity after exposure to proteases and found that the cubosomes did not display cytotoxicity against keratinocytes. Furthermore, the cubosomes were effective in an ex vivo porcine skin wound infection model. The authors did note that due to strong association with the cubosomes, almost no release of LL-37 was observed after 24 h; therefore, they suggested formulations allowing triggered release of the AMP might mitigate the observed loss of antimicrobial activity and allow for increased local peptide concentration. Similarly, nanostructured lipid carriers (NLCs) and solid lipid nanoparticles (SLNs) have been studied for their use as liposome alternatives for AMP carriers due to their improved stability, shelf life, encapsulation efficiency and scalability [23]. For example, LL37 was encapsulated in a NLC for topical wound treatment, where it maintained its immunomodulatory and antimicrobial activity while improving wound healing compared to the free peptide [104]. This is mirrored by work performed by Fumakia and Ho, who found that SLN formulations containing LL37 and the elastase inhibitor Serpin A1 accelerated wound healing and displayed synergistic antimicrobial and anti-inflammatory activities [105]. Sans-Serramitjana et al. also found that encapsulating colistin with NLC maintained the peptide’s antimicrobial activity, which was more stable over time compared to SLN-encapsulated colistin [106].

### 3.2. Metal Nanoparticles

Metal nanoparticles possess unique physical, electronic, and magnetic properties that make them useful for a variety of biomedical applications. Furthermore, their large surface area-to-volume ratio enables a large multitude of molecules to be loaded onto their surface [77], making them particularly attractive as drug delivery vehicles. In fact, many metal nanoparticles themselves have been shown to disrupt the bacterial cell membrane and react with intracellular targets, thereby providing some of their own antimicrobial activity that could compound with the activity of bound AMPs [77]. However, cytotoxicity of metal nanoparticles tends to be high, especially at smaller sizes, and there remain concerns regarding unknown interactions with cellular functions, insufficient clearance from the body, and generation of ROS [77].

#### 3.2.1. Gold Nanoparticles

In contrast to other metal nanoparticles, gold nanoparticles (AuNPs) generally show high biocompatibility and non-immunogenicity, as gold is an inert, nontoxic element [77]. Furthermore, gold itself is able to reduce ROS, mitigating those formed during administration of nanoparticles [77]. However, the toxicity of AuNPs still depends heavily on their size and functionalization and thus requires individual optimization.

Nevertheless, AuNPs have been harnessed to improve the stability and activity of bound AMPs [107]. AuNPs can be functionalized with a number of moieties for covalent conjugation (e.g., *N*-hydroxysuccinimide esters, maleimides, alkynes/azides), but the majority of AMPs are conjugated directly using a metal thiolate Au-S bond via an incorporated *N*- or *C*-terminal cysteine residue [77]. This presents a simple method for conjugation within aqueous buffer. Site-specific conjugation using cysteine is particularly useful in lysine-containing AMPs, as their amine groups can interact with the AuNP and potentially affect activity by holding the peptide close to the AuNP core, thereby preventing sufficient access to bacterial membranes [108]. Conjugation to AuNPs has been shown to markedly enhance the proteolytic stability of AMPs against trypsin while minimally affecting their in vitro activity, thereby improving their active lifetimes [108]. Furthermore, Chowdhury et al. found that AuNP-AMPs displayed good biocompatibility both in vitro and in vivo and were able to internalize into epithelial and macrophage cells without toxicity to destroy internalized *Salmonella* pathogens, a feat also mirrored in an in vivo mouse model [109]. Additionally, the conjugation of AMPs to AuNPs has been utilized to improve the activity of the bound peptide. Recently, Zheng et al. conjugated daptomycin to mercaptopyrimidine-functionalized gold nanoclusters shown to destroy bacteria via ROS production, membrane damage and DNA destruction, to produce a synergistic antimicrobial effect [110]. Palmieri et al. also found that conjugating a small synthetic AMP to AuNPs increased its killing ability against both Gram-positive and Gram-negative bacteria at a concentration less than 100 nM [111].

Interestingly, AMP-AuNPs utilizing DNA aptamers for targeting have been shown to be very effective against intracellular bacteria [37,38]. These aptamers allow specific targeting with high binding affinity and long-term stability, and the properties of the DNA oligonucleotides impart efficient cellular uptake of the conjugated nanoparticles [112]. In both cases, the conjugates displayed low toxicity in vitro and strong in vitro and in vivo antimicrobial activity against intracellular bacteria, with the conjugates conferring complete subject survival compared to the complete lack thereof in untreated mice [37,38]. The authors suggested the aptamers imparted an increased cellular uptake of the conjugates, providing improved delivery for targeting intracellular pathogens.

#### 3.2.2. Silver Nanoparticles

One of the most popular metals used in nanoparticle synthesis, silver is an attractive delivery system for AMPs due to its own innate antimicrobial activity against both Gram-positive and Gram-negative bacteria [23,113]. Like AuNPs, AgNPs are small in size, have high surface area-to-volume ratios, and display unique electrical and optical properties resulting in high reactivity and chemical stability [23]. In addition, thiol chemistry can be utilized for direct conjugation to the AgNP, which allows for an easy method of conjugation that has been shown to increase both the stability and activity of the bound AMP [114,115,116]. However, AgNPs generally exhibit high cytotoxicity and are thus not necessarily a great delivery vehicle for improving AMP tolerability. In fact, researchers have utilized surface peptide conjugation to mitigate the cytotoxicity of AgNPs [117,118]. For example, Gao et al. conjugated AgNPs with short peptides, which significantly reduced the cytotoxicity of the AgNPs to levels similar to those of the free peptide while also improving their antimicrobial activity [118].

Nevertheless, many researchers have combined the antimicrobial activities of AgNPs and AMPs for enhanced activity [116,117,118,119]. Recently, Pal et al. conjugated the AMP Andersonin-Y1 to AgNPs by adding a cysteine on the *N*- or *C*-terminus of the peptide [119]. The conjugates, which contained nearly 200 peptides per AgNP, increased the activity of the non-cysteine AMP nearly 10-fold, though this activity was comparable to the free cysteine-containing peptides. Interestingly, the hemolytic activity of both the AMPs and the AgNP-AMP conjugates was low, indicating good biocompatibility.

### 3.3. Synthetic Polymers

Synthetic polymers have been extensively employed for drug delivery, with some conjugates being the first formulations translated into clinical practice. In general, synthetic water-soluble polymers impart improved solubility, enhanced biocompatibility and stability, and prolonged circulation to the payload [26]. For AMP delivery, multiple polymeric structures have been studied, including linear polymers (e.g., PEG), dendrimeric and dendrimer-like polymers (e.g., hyberbranched polyglycerol, HPG), and polymeric nanoparticles (e.g., poly(lactide-co-glycolide), PLGA). Perhaps the most well-known and well-studied polymer, PEG, imparts its biocompatibility on conjugated drugs by steric repulsion, which shields antigenic epitopes and prevents proteolytic enzyme degradation and opsonisation [26]. Other water-soluble polymers likely improve the biocompatibility of their therapeutic agents in the same manner.

PEGylation is one of the most extensively utilized methods to improve the biocompatibility of drugs, garnering the most success in clinical trials thus far [26]. It has been used to drastically reduce the cytotoxicity of a variety of AMPs, including the cyclic AMP tachyplesin I [120], a magainin 2 analogue [121], the heparin cofactor II fragment KYE28 [122] and the synthetic AMP CaLL [123], but in each case listed, PEGylation also resulted in significant decreases in antimicrobial activity. However, both the improved biocompatibility and decreased activity of PEGylated AMPs have been found to be dependent on PEG length, so optimization of PEG length can potentially be used to improve the selectivity toward bacteria in blood [122]. Nevertheless, the reduced activity associated with PEGylation coupled to the limitations of PEG itself, including non-biodegradability, high intrinsic viscosity, large hydrodynamic size, and singular functionality [3], has limited the use of PEG as an AMP delivery vehicle.

As mentioned earlier, PLGA nanoparticles have been used extensively for the encapsulation of AMPs, as they allow prolonged drug release as well as enhanced stability by protecting the AMP against the biological environment. Furthermore, PLGA itself has been found to enhance wound healing, which makes it particularly appropriate for wound treatment, where AMPs are often used [124], and is biodegradable, minimizing the worries regarding bioaccumulation [125]. For AMP delivery, it has been shown to improve the efficacy of the AMP plectasin against *S. aureus*-infected bronchial epithelial cell monolayers [126], and to fully protect fish from pathogenic bacteria by providing prolonged release of the AMP pleurocidin [127]. Furthermore, as mentioned earlier, pH-responsive PLGA nanoparticles functionalized with poly-L-histidine have been used to improve delivery of encapsulated vancomycin to bacteria [42]. AMP-conjugated PLGA nanoparticles entrapping growth factors has also been investigated for the co-delivery of these two therapeutic agents for wound healing, displaying moderate broad-spectrum antimicrobial activity with sustained growth factor release [128]. Recently, Casciaro et al. showed that encapsulation of esculentin-1 AMP derivatives in PLGA nanoparticles coated with poly(vinyl alcohol) to prevent aggregation improved their transport through an artificial lung mucus and bacterial barrier [129]. Though the peptide’s in vitro activity decreased upon encapsulation, it displayed 4–17-fold enhancement of activity in an in vivo lung infection mouse model, suggesting these PLGA-AMP nanoparticles could have strong potential for the treatment of bacterial pathogens in cystic fibrosis patients.

Our labs have also worked heavily on the use of hyperbranched polyglycerol (HPG) for the delivery of AMPs. HPG is a dendrimer-like hyperbranched polymer that has garnered attention due it its high biocompatibility, stability, tunability, and multi-functionality [78]. Furthermore, HPG synthesis is simple, and the polymer can be made biodegradable by, e.g., incorporating acid-sensitive moieties, to prevent in vivo bioaccumulation. Thus, HPG has been studied extensively for a wide variety of biomedical applications ranging from cell surface engineering and organ preservation to macromolecular therapeutics and drug delivery. An in-depth review of HPG and its applications can be found in [78].

Our groups demonstrated that conjugating the AMP aurein 2.2 to a moderate molecular weight HPG significantly improved its biocompatibility in vitro in both blood and cell culture [130]. Unfortunately, however, the conjugation also decreased the peptide’s activity against *S. aureus* and *Staphylococcus epidermidis*, though to less of an extent than those exhibited by some PEGylated AMPs [130]. It is thought that this decrease could be due to interactions of the peptides with the polymer or changes in peptide folding. Interestingly, both the biocompatibility and antimicrobial activity of the conjugates depended on their peptide density, which in this case ranged from 7–18 peptides per polymer, highlighting the promising capacity to tune HPG conjugates for specific drug delivery. We further studied the tunability of HPG by conjugating an aurein 2.2-derivative with improved antimicrobial activity to HPG of different molecular weights [131]. Once again, the conjugates displayed excellent biocompatibility and also exhibited resistance to trypsin degradation. Though the activity of the peptide did decrease, the most promising conjugate, a low molecular weight HPG containing 7–8 conjugated peptides, still displayed moderate activity in vitro against *S. aureus*. Similarly, Haney et al. found that derivatized HPGs containing a hydrophobic polymer core capped with carboxylic acid-functionalized PEG groups significantly reduced the aggregation of the synthetic AMP IDR-1018 both in vitro and in vivo while maintaining the immunomodulatory activity of the peptide [132]. This is of particular interest as peptide aggregation has been shown to inhibit the activity of certain AMPs and is thought to be partially responsible for AMP toxicity [99,132]. Notably, this formulation did not require covalent conjugation or modification of the parent peptide and was more efficacious than other tested formulations utilizing hyaluronic acid, carboxymethyl cellulose, and hydroxypropyl methyl cellulose.

### 3.4. Natural Biomolecules

Concerns regarding the potential immunogenicity and non-biodegradability of synthetic polymers have generated interest in utilizing natural biomolecules for AMP delivery. These delivery vehicles include natural polymers such as polysaccharides, polypeptides and antibodies, and DNA nanostructures.

#### 3.4.1. Polysaccharides

Being a linear polysaccharide made up of glucosamine and *N*-acetylglucosamine subunits [133], chitosan is a biocompatible, non-toxic and biodegradable polysaccharide that also displays bioadhesive and wound healing properties [79]. Furthermore, it itself displays mild antibacterial properties against a broad spectrum of microorganisms, which can be tuned by molecular weight and chemical modification via amine groups (e.g., deacetylation) [79]; as such, it has gained some attention as a promising delivery vehicle for AMPs. Much work has been published detailing the utilization of chitosan to improve the biocompatibility of peptides while maintaining or enhancing their antimicrobial activity [79,133,134,135,136]. For example, Hou et al. developed self-assembled short chain chitosan-polylysine AMP nanoparticles that displayed strong broad-spectrum activity both in vitro and in vivo with very low hemolytic activity and improved selectivity [136]. In a similar fashion, cyclic oligosaccharide cyclodextrins have been harnessed for AMP delivery [137,138,139]. Not only have these formulations displayed high stability, solubility, and protease protection, but they also maintained good antimicrobial activity.

#### 3.4.2. Polypeptides 

Cell penetrating peptides (CPPs) themselves have been harnessed to improve the antimicrobial activity of AMPs. Interestingly, these CPPs often negligibly affect the cytotoxicity of the AMPs while improving their antimicrobial activity and specificity, particularly when targeting Gram-negative bacteria [140,141]. Hydrophilic polypeptides (e.g., XTEN and PAS) have also gained much interest in recent years as alternatives to polymers for protein conjugation [26], though no examples of AMP conjugation have yet been published. Recently, short polycationic hexa-arginine polypeptides were conjugated to vancomycin to bypass multiple modes of vancomycin resistance, accompanied by a 1000-fold increase in antibacterial activity [142]. The peptide conjugates demonstrated efficacy in vivo along with improved biodistribution and excretion profiles compared to free vancomycin. Interestingly, the conjugates displayed no hemolytic activity or cell culture toxicity. Additionally, it has been shown that certain AMPs can be structurally manipulated to form dendrimers by introducing branching peptide chains via reactive amino acid side chain groups (e.g., lysine amines) [82]. These dendritic structures have displayed strong antimicrobial activity alongside improved serum stability and reduced hemolysis [82]. Likewise, tetrabranched AMP oligodendrimers, also termed multiple antigen peptides, have exhibited strong in vitro and in vivo activity as well as stability against blood proteases. For example, the branched AMP M33-L displayed strong activity against a wide variety of bacteria, and also exhibited LPS neutralization, serum stability and efficacy in mouse sepsis models [143,144]. Extensive in vivo studies of M33-L also showed the peptide to be much less toxic than the clinically used peptide colistin [144]. Isomerization of the oligodendrimer further improved the AMP’s activity against Gram-positive bacteria, potentially due to increased resistance to bacterial proteases such as staphylococcal aureolysin and elastase [145].

#### 3.4.3. Antibodies

Though research exploring antibodies as AMP delivery vehicles is limited, their efficacy against both extracellular and intracellular drug resistant bacteria as antibody-antibiotic conjugates (AACs) in recent years suggests they could behave as an attractive delivery vehicle for AMPs. On top of potential pathogenic specificity that prevents native microbiome disruption, antibodies exhibit long half-lives, slow clearance, and great biocompatibility [60], making them promising drug delivery vehicles. For example, Lehar et al. conjugated a rifamycin analogue to an antibody that binds specifically to the β-*N*-acetylglucosamine residues of cell-wall teichoic acids (WTAs) on Gram-positive bacteria [56]. This antigen was selected due to its high abundance in *S. aureus* cell walls and due to the fact the enzyme responsible for its incorporation into WTAs is conserved throughout all multidrug resistant methicillin-resistant *S. aureus* (MRSA) strains owing to its role in β-lactam antibiotic resistance [60]. The two components were connected by a cathepsin-cleavable dipeptide for specific release inside the phagolysosome upon uptake of opsonized *S. aureus*. The conjugate showed strong activity in a mouse bacteraemia model against MRSA, while simultaneously providing evidence for the role of intracellular *S. aureus* in invasive infection.

In regards to antibody-AMP conjugation, Franzma et al. conjugated the AMP SMAP28 to rabbit immunoglobulin (IgG) antibodies specific to the outer surface of a strain of *Porphyromonas gingivalis* [146]. They found that the IgG-SMAP28 conjugate showed strong activity against the targeted strain, but its specificity was limited, particularly at higher conjugate concentrations. In a similar vein, antibody–antifungal peptide fusion proteins specific to the cell surface of *Fusarium* fungi conferred protection against the fungal pathogen when expressed in a plant model [147].

More recently, Touti et al. developed antibody-macrocylic peptide conjugates specific for *E. coli* [148]. The cathelicidin-derived AMPs were macrocyclized using cysteine arylation macrocylization chemistry to protect against serum degradation, while the antibody targeted core lipopolysaccharide glycans that are present in high density on the surface of the bacteria. The peptides were conjugated enzymatically to the antibody using sortase A via acceptor and donor peptide tags on the antibody and AMP, respectively. The most promising conjugates displayed strong bactericidal activity against *E. coli* with negligible hemotoxicity, though once again, the specificity was diminished at higher concentrations. It should be noted that half of the studied peptides lost their activity upon conjugation, highlighting the challenges that could arise in the formulation of non-releasing antibody–AMP conjugates.

As with all carriers, antibodies have some drawbacks for clinical translation. Firstly, their high specificity (i.e., narrow spectrum), at times advantageous, can also present a disadvantage in the broad-spectrum treatment of bacteria and polymicrobial infection. Furthermore, their specificity requires extensive optimization, as detailed by the examples given earlier, which can be costly [84]. And secondly, their targeting ability has the potential to be thwarted by cell-wall components, such as wall techoic acids in Gram-positive bacteria, which can conceal surface epitopes [149], potentially enabling the development of resistance mechanisms. Nevertheless, the large number of human monoclonal antibody (mAb) therapies currently being developed and tested in clinical trials to treat bacterial infection, either by direct pathogen targeting or by toxin neutralization [84], highlights the strong potential of these molecules for AMP delivery.

#### 3.4.4. DNA Nanostructures

DNA nanostructures have gained much attention in recent years for a wide array of applications, but they are particularly appealing for drug delivery due to their small size, high solubility, high biocompatibility, biodegradability, responsiveness to stimuli, and their suitability for precise attachment of therapeutic agents [85,150]. Recent developments in their formulation and characterization have suggested they could be promising delivery vehicles for AMPs. For example, DNA origami nanostructures functionalized with aptamers targeting both Gram-positive and Gram-negative bacteria have been used as vehicles for delivering lysozymes to target infection sites [151]. Recently, Obuobi et al. developed DNA nanostructure hydrogels that released a broad spectrum AMP upon degradation by nuclease-secreting MRSA [150]. The abundant anionic phosphate groups of the DNA nanostructures allowed for encapsulation of the cationic peptide via strong electrostatic interactions, thereafter yielding spontaneous formation of a hydrogel. The AMP-loaded nanostructures demonstrated sustained local release of the peptide in response to environmental nucleases, significantly reducing the toxicity of the peptide against mammalian dermal cell lines. Furthermore, the hydrogels maintained activity against MRSA in vitro, and a single application of the hydrogel in an in vivo porcine skin model resulted in significant bacterial reduction. Interestingly, the researchers also noted anti-inflammatory properties of the hydrogel, which improved wound healing rates in vivo. These properties are thought to arise from the DNA nanostructures themselves, highlighting another potential advantage of using these structures for AMP delivery.

## 4. Alternatives: Internal Modification of AMPs

Though this review focuses on delivery vehicles for AMP administration, it is also necessary to discuss the strategies being developed to modify the peptide itself for increased biocompatibility, stability, activity and specificity. These chemical modifications include incorporation of d-amino acids, lipidation, rational amino acid substitution, cyclization/stapling and peptidomimetics.

Firstly, replacement of l-amino acids for d-amino acids is a commonly utilized approach to protect the AMP from stereospecific degradation by proteases [3]. The alteration of amino acid stereochemistry often does not affect the antimicrobial activity of the peptide and can be used to prolong the activity in the presence of proteases. For example, Jia et al. found that the AMP polybia-CP could be made resistant to trypsin and chymotrypsin degradation by replacing all l-amino acids for d-amino acids without compromising its antimicrobial activity against both Gram-positive and Gram-negative bacteria [152]. Interestingly, they also noted that a single substitution of the peptide’s l-lysine with d-lysine only slightly decreased the AMP’s antimicrobial activity while both improving the peptide’s stability and substantially decreasing its hemolytic activity [153]. It should be noted that d-peptide synthesis is quite costly [154], which hinders its straightforward translation into clinical practice.

By attaching fatty acid chains to the amine groups of lysine or *N*-terminus residues, lipidation can be harnessed to improve the activity of AMPs by improving their hydrophobic interactions with bacterial membranes [155]. Recently, Kamysz et al. found that *N*-terminus conjugation of C_4_-C_14_ fatty acid chains enhanced the activity of an alpha-helical LL-37 fragment against a variety of MDR pathogens [156]. The activity varied with fatty acid chain length, with C_8_ chains displaying the highest activity and larger C_14_ chains displaying lower activity due to self-assembly into large aggregates. Unfortunately, however, lipidation increased the cytotoxicity of the peptides proportional to carbon length [156], which is consistent with other reports [157]. Both the improved activity and increased aggregation of higher carbon chain tails agree with previous work on *N*-terminal lipidation [158]. In 2018, Siriwardena et al. developed C_10_-lipidated AMP dendrimers that showed improved broad-spectrum antimicrobial activity against MDR bacteria in vivo, though once again, hemolysis increased with lipid length [157]. Interestingly, Lombardi et al. harnessed the proclivity of these lipidated AMPs to self-assemble to generate nanostructures with improved antibiofilm activity and protease stability [155]. This was accomplished by linking the AMP to an aliphatic polyalanine peptide attached to a C_19_ lipidic tail.

There has been a great interest in recent years to determine qualitative design principles that could be utilized to strategically design AMPs with improved antimicrobial activity and reduced hemolytic activity [154,159,160]. To generate this specificity, the positioning of positively charged amino acids—in particular, lysine—within the hydrophilic and hydrophobic segments of the AMP seems to be especially important [154,159]. For example, substitution of lysine in the hydrophilic face of an AMP has been shown to improve its antimicrobial activity against both Gram-positive and Gram-negative bacteria with slight reductions in hemolytic activity; conversely, substitution in the hydrophobic face decreases hemolysis but has variable consequences in activity, the extent of which depends on the residue being replaced [154].

Cyclization of AMPs is a common method utilized to protect the peptide against protease degradation. This can be accomplished by utilizing disulphide bridges via cysteine residues akin to human defensins [161] or, more frequently, by incorporating one or more staples between i, i + 4 or i + 7 residues, which lock the peptide into an alpha-helical structure with twisted amide bonds that are less favourable to protease degradation [154,162]. A number of techniques have been developed to form these staples, including sulphur and nitrogen arylation of cysteine and lysine residues, respectively [163,164]. Though staples do indeed increase proteolytic resistance and alpha-helicity, these properties are often accompanied by unpredictable and undesired effects on antimicrobial activity, mammalian membrane lysis, and peptide solubility [154,162]. Recently, however, Mourtada and colleagues studied the structure-function-toxicity relationship of 58 stapled AMP (StAMP) constructs of magainin II to devise an algorithm for the in silico design of stable, nontoxic StAMPs [154]. By sequentially incorporating i, i + 4 staples at different positions in the magainin II sequence, they determined that staple placement had minimal undesired effects on hemolytic activity when placed within an already established hydrophobic face; conversely, placement in a region of low hydrophobicity that expanded the entire hydrophobic space markedly increased the activity. Through their designed algorithm, they were then able to generate three additional StAMPs with strong antimicrobial activity and little to no hemolysis without having to generate a potentially costly StAMP library.

Finally, we would be remiss to exclude the peptidomimetic approaches that have shown great potential as alternatives to AMPs. These peptide mimics imitate the structure, activity and mode of action of AMPs by conserving the overall amphiphilic structures of the peptides but changing the chemical composition of the backbone, which is often done with modified amino acids or amino acid-like units [3,165,166]. Consequently, peptidomimetics are able to improve in vivo half-life and stability by removing the peptide’s inherent protease susceptibility, and can also improve toxicity and synthesis costs [165,166,167,168]. Indeed, a number of these compounds have been successful in clinical trials, such as the defensin mimetic brilacidin [168,169]. Notably, Luther et al. recently developed a class of peptidomimetics consisting of a mixture of natural amino acids and synthetic building blocks arranged into two linked macrocycles, one of which was derived from the polymixin and colistin peptides [170]. A number of the derived compounds exhibited good activity against a wide array of MDR pathogens, low cell toxicity, maintenance of activity in complex media such as human serum, and strong efficacy in several mouse models of infection alongside favourable tolerability and pharmacokinetics. As the field of peptidomimetics is quite large and can be categorized into a number of families, we refer you to in-depth reviews by Molchanova et al. [165], Ghosh et al. [166], and Kuppusamy et al. [168] for more information.

## 5. Conclusions

AMPs provide a promising solution to the ever more pressing issue of bacterial multidrug resistance. However, many hurdles block the successful translation of AMPs into clinical practice, as they often display high toxicity, low stability and rapid clearance in biological systems. Moreover, AMPs tend to display less direct antimicrobial activity compared to conventional antibiotics, which immediately places them at a disadvantage, as regulatory agencies require that novel antimicrobial agents exhibit similar or stronger activity, i.e., non-inferiority or superiority, compared to those currently available [1,171,172]. As such, it is imperative to develop strategies to formulate these AMPs without hampering their innate, often multimodal activities. In this regard, a multilayered approach combining delivery vehicles, peptide modification, and infection site release strategies may provide the best solution. Harnessing these strategies will hopefully place these promising anti-infective compounds at the forefront of infectious disease treatment in the near future.

## Figures and Tables

**Figure 1 molecules-25-03048-f001:**
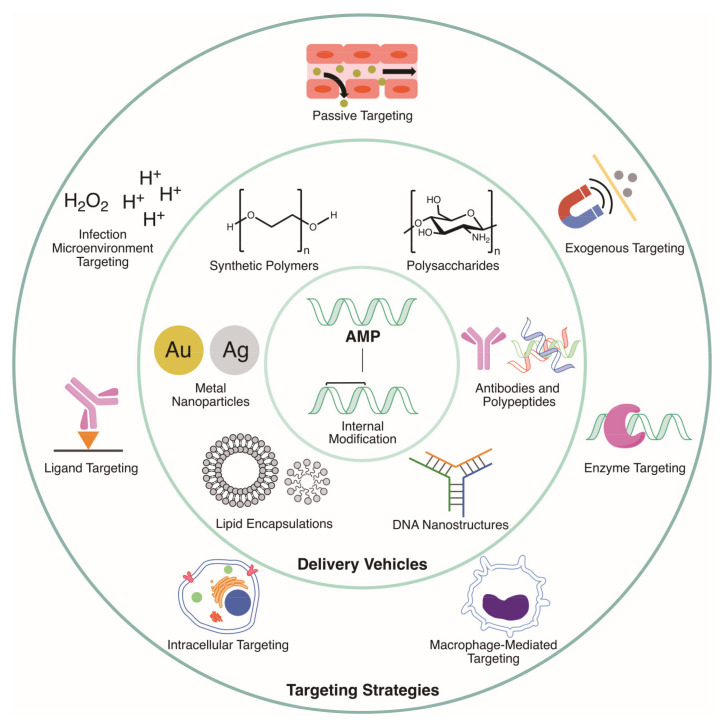
Summary of all of the targeting strategies, delivery vehicles, and AMP modifications presented in this review.

**Table 1 molecules-25-03048-t001:** Summary of the delivery vehicles discussed herein, as well as some of their advantages and disadvantages.

Delivery Vehicle	Advantages	Disadvantages	References
Liposomes	Biodegradable Low toxicity and immunogenicity Easy surface modification Applicable to hydrophobic and hydrophilic content Protect encapsulated peptide from proteolytic degradation Can passively target infection site via the EPR effect	Low stability in vivo due to phagocytic clearance Limited sterilization techniques for scaling up Poor drug loading capacity Limited control over site-specific drug release Can elicit pseudoallergic response during intravenous delivery	[22,23,70,71]
Polymeric Micelles	Higher stability and loading capacity than liposomes Long circulation by reducing uptake by the RES Responsive to stimuli Small in size, which can improve target site accumulation by increasing tissue penetration and distribution	Content leakage Low stability upon environmental change (e.g., upon blood injection) resulting in premature drug release and potential off-site toxicity	[72,73,74,75]
Liquid Crystalline Nanoparticles	Biodegradable Thermally stable	Low encapsulation efficiency Poor shelf-life	[22,23]
Nano-structured Lipid Carriers and Solid Lipid Nanoparticles	Biodegradable Biocompatible Improved stability and shelf-life Improved encapsulation efficiency Feasible large-scale manufacturing Suitable for a variety of administration routes (e.g., oral, parenteral, ocular)	Low encapsulation efficiency of polar content Site-specific drug release not explored Premature drug expulsion in storage and/or upon administration	[23,71,76]
Metal Nanoparticles	Innate antimicrobial activity High surface area-to-volume ratio, allowing for high loading of multiple molecules (i.e., multimodal) Easy functionalization Magnetic, electrical, and thermal properties can enable exogenous targeting	High toxicity Unknown interactions with cellular functions Poor clearance and non-biodegradable Poor stability Poor shelf-life Properties highly dependent on size, requiring individual characterization Size disparity between repeat syntheses	[23,77]
Synthetic polymers	High biocompatibility Easy surface modification for enhanced targeting and multi-functionality (e.g., dendritic polymers) High solubility in water High drug payload Easy to include controlled drug release	Potential immunogenicity Non-biodegradable, except in specific cases Costly multistep synthesis when functionalized polymers are used Low cell affinity	[22,70,78]
Poly-saccharides	High biocompatibility Low toxicity Biodegradable Often possess innate bioactive properties (e.g., wound healing, bioadhesion, antimicrobial activity) High solubility in water	Properties can vary significantly based on source Chemical modification often required for uniform physiochemical properties Unpredictable degradation profiles, particularly when chemically modified	[79,80,81]
Polypeptide Dendrimers	Biodegradable Resistant to proteolytic degradation due to steric hindrance Increased antimicrobial activity	High production cost Laborious synthesis Potential immunogenicity of larger dendrimers	[82,83]
Antibodies	High specificity, preventing disruption of microbiome High biocompatibility High stability Long half-life Innate targeting for site-specific drug delivery	Extensive optimization required High development and scaling-up costs Specificity may be non-ideal for polymicrobial infections	[84]
DNA Nano-structures	Biodegradable High solubility in water Precise molecular attachment possible Able to evade immune system Stimuli-responsive, allowing for spatiotemporal control of delivery	Susceptible to nuclease degradation if not chemically modified Challenging and costly large- scale production Complex, if functionality is required Off-target effects unknown	[85]

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
