# Peer review of "Towards Robust Delivery of Antimicrobial Peptides to Combat Bacterial Resistance"

_molecules, 2020, doi:10.3390/molecules25133048_

Round 1

Reviewer 1 Report

Prof. Suzana K. Straus and the colleagues exhibit antimicrobial peptide delivery systems to express the function in this review article. This review is very suitable for the Molecules

Author Response

Prof. Suzana K. Straus and the colleagues exhibit antimicrobial peptide delivery

systems to express the function in this review article. This review is very suitable for the Molecules.

We thank Reviewer 1 for their positive comments.

Reviewer 2 Report

The review by Drayton and co-workers provides a nice compendium of potential methods to improve AMP delivery. It is well written, methodically organised and provides a detailed account on the current state of the art.

However, the viewpoint is unbalanced. Here, it could benefit by putting the perceived need for advanced delivery systems into perspective by considering the population of AMPs already in clinical trials (eg. DOI: 10.1002/pep2.24122) that evidently do not need an advanced delivery system. Most of these AMPs are delivered topically - do they suffer from known issues that would necessarily require or benefit from site-specific delivery ? Are targeting and release modalities likely to be for systemically administered therapies more so than topical ones? 

The exact definition of AMPs, as it pertains to this review, should also be emphasised in the introduction. Given that AMPs can cover a broad suite of chemo types (polymyxins, daptomycin, teixobactin, murepavidin, phage lysins like exebacase etc etc), and then of course all of the peptidomimetic variants, it would help the reader focus on the subset of AMPs that are likely to benefit from delivery approaches.  

Further to this, when one considers the current antimicrobial pipeline, and the harsh realities of developing non-traditional therapies, AMPs don't appear to rank very highly in terms of potential for translation outcomes, and obviously technologies for better delivery are desirable (hence the nature of this manuscript). It would be good to put the field of AMP into perspective by mentioning this, eg. DOI:10.1016/j.chom.2019.06.004, DOI:10.1016/ S1473-3099(15)00466-1.

Therefore, the reader would greatly benefit from understanding the current state of the field and what gaps are likely to be filled from advanced delivery systems, before diving into the rest of the article. 

I think the review would also read better if the approaches outlined in sections 2 & 3 were summarily tabulated to provide a at-a-glance summary of each method, pros vs cons etc. It would also help the reader to compare and contrast, and solidify a take home message.

In section 4, the major theme missing is peptidomimetic approaches. I realise this is not the focus of the review, but if the authors are to go there, then one must highlight this given some are in clinical trials, eg. brilacidin. 

Lastly, line 570 contains a sweeping statement about the regulatory need for "stronger activity". The developmental pathway for most antimicrobials will be demonstration of non-inferiority. The authors should clarify this sentence. (eg. DOI:10.1038/s41467-019-11303-9)

Minor comments:

  1. I wonder if the authors should reflect on the use of the word "robust" in the title?
  2. line 217: this study here (DOI: 10.3390/antibiotics9020094) by the team at combioxin is a nice example of lipid decoys that might complement the review
  3. line 259: it would be useful to mention the route of administration is SC by intra abscess injection
  4. there are occasional non-italicised use of species names eg. line 397, 398 etc. please ensure consistency throughout
  5. I noticed that when referencing authors specifically, the reference is given as a date, whereas others are given specific numbers. this seems a little unusual
  6. lines 479-485: DOI 10.3390/antibiotics9040155 might be another useful reference here
  7. line 511: use of L- and D- notation should be small CAPS
  8. line 522-536 - seems to be repetition with references (eg. ref 139 and 142) which could be consolidated 

Author Response

Reviewer #2: The review by Drayton and co-workers provides a nice compendium of potential methods to improve AMP delivery. It is well written, methodically organised and provides a detailed account on the current state of the art.

2 We thank Reviewer 2 for their positive comments.

However, the viewpoint is unbalanced. Here, it could benefit by putting the perceived need for advanced delivery systems into perspective by considering the population of AMPs already in clinical trials (eg. DOI: 10.1002/pep2.24122) that evidently do not need an advanced delivery system. Most of these AMPs are delivered topically - do they suffer from known issues that would

necessarily require or benefit from site-specific delivery ? Are targeting and release modalities likely to be for systemically administered therapies more so than topical ones?

We agree and have added a paragraph at the end of the introduction. It reads: “This review does not focus on specific AMP classes or subtypes, but rather examines a range of delivery vehicles applied to a broad definition of AMPs (i.e., linear peptides, lipopeptides, glycopeptides). The review also presents alternatives to delivery vehicles (i.e., D-amino acid substitution, lipidation, rational amino acid substitution, cyclization/stapling and peptidomimetics). Further, as the majority of AMPs being tested clinically are for topical administration where concerns regarding degradation, toxicity and specificity are lessened [24,25], we focus mainly on systemically administered therapies that would benefit most from these advanced delivery systems, though some topical systems are also mentioned. Indeed, AMPs are considered viable alternatives to conventional antibiotics and have gathered significant support for clinical evaluation, but the difficulties associated with systemic therapy necessitate additional formulation designs to improve stability and efficacy and to decrease toxicity [25]. By examining the various strategies that are already available, this review aims to inspire the future development of more robust delivery strategies.”. This also incorporates the reference mentioned.

The exact definition of AMPs, as it pertains to this review, should also be emphasised in the introduction. Given that AMPs can cover a broad suite of chemo types (polymyxins, daptomycin, teixobactin, murepavidin, phage lysins like exebacase etc etc), and then of course all of the peptidomimetic variants, it would help the reader focus on the subset of AMPs that are likely to benefit from delivery approaches.

We agree and have defined what we mean by AMPs in the context of this review (see above paragraph). We have also added a paragraph on peptidomimetics in section 4. 

Further to this, when one considers the current antimicrobial pipeline, and the harsh realities of developing non-traditional therapies, AMPs don't appear to rank very highly in terms of potential for translation outcomes, and obviously technologies for better delivery are desirable (hence the nature of this manuscript). It would be good to put the field of AMP into perspective

by mentioning this, eg. DOI:10.1016/j.chom.2019.06.004, DOI:10.1016/ S1473-3099(15)00466-

1.

Again, we agree and have added sentences which hopefully capture this (see paragraph in the point above). The first reference has been incorporated in the text in the conclusions, and the second is reference [25] listed in the paragraph above.

Therefore, the reader would greatly benefit from understanding the current state of the field and

what gaps are likely to be filled from advanced delivery systems, before diving into the rest of the 3 article.

We thank the reviewer for these suggestions and agree that clearly defining the current state of the field is important before proceeding to describe what options are available in the rest of the article. We hope that the changes made are satisfactory.

I think the review would also read better if the approaches outlined in sections 2 & 3 were summarily tabulated to provide a at-a-glance summary of each method, pros vs cons etc. It would also help the reader to compare and contrast, and solidify a take home message.

We have included a table which lists the different delivery vehicles presented in the body of the text and lists advantages and disadvantages (pages 7-8). We have not reproduced it in the response to save space.

In section 4, the major theme missing is peptidomimetic approaches. I realise this is not the focus of the review, but if the authors are to go there, then one must highlight this given some are in clinical trials, eg. brilacidin.

We agree and hence have included a new paragraph at the end of section 4. It reads: “Finally, we would be remiss to exclude the peptidomimetic approaches that have shown great potential as alternatives to AMPs. These peptide mimics imitate the structure, activity and mode of action of AMPs by conserving the overall amphiphilic structures of the peptides but changing the chemical composition of the backbone, which is often done with modified amino acids or amino acid-like units [3,165,166]. Consequently, peptidomimetics are able to improve in vivo half-life and stability by removing the peptide’s inherent protease susceptibility, and can also improve toxicity and synthesis costs [165–168]. Indeed, a number of these compounds have been successful in clinical trials, such as the defensin mimetic brilacidin [168,169]. Notably, Luther et

  1. recently developed a class of peptidomimetics consisting of a mixture of natural amino acids and synthetic building blocks arranged into two linked macrocycles, one of which was derived from the polymixin and colistin peptides [170]. A number of the derived compounds exhibited good activity against a wide array of MDR pathogens, low cell toxicity,  maintenance of activity in complex media such as human serum, and strong efficacy in several mouse models of infection alongside favourable tolerability and pharmacokinetics. As the field of peptidomimetics is quite large and can be categorized into a number of families, we refer you to in-depth reviews by Molchanova et al. [165], Ghosh et al. [166], and Kuppusamy et al. [168] for more information.”

Lastly, line 570 contains a sweeping statement about the regulatory need for "stronger activity". The developmental pathway for most antimicrobials will be demonstration of non-inferiority. The authors should clarify this sentence. (eg. DOI:10.1038/s41467-019-11303-9)

We have adopted the language used in the reference given by Reviewer 2. The sentence now reads: “Moreover, AMPs tend to display less direct antimicrobial activity compared to conventional antibiotics, which immediately places them at a disadvantage, as regulatory agencies require that novel antimicrobial agents exhibit similar or stronger activity—i.e., noninferiority or superiority—compared to those currently available [1,171,172].”

4 Minor comments:

  1. I wonder if the authors should reflect on the use of the word "robust" in the title?

Given the sentence added at the end of the introduction (see above), we would like to keep the term “robust”.

  1. line 217: this study here (DOI: 10.3390/antibiotics9020094) by the team at combioxin isa nice example of lipid decoys that might complement the review

This has been added to the text. Lines 250-255 (numbering changed due to edits) read: “An example of these agents is CAL02, a clinically tested liposomal antivirulence drug that entraps and neutralizes a large array of virulence factors produced by both Grampositive and Gram-negative bacteria [71]. By mimicking the cell membrane lipid composition that many virulence effectors target, these liposomes can bind to these toxins with higher affinity than cells, thereby preventing the threatening complications these molecules are associated with during severe infections (e.g., tissue damage and organ failure).”

3. line 259: it would be useful to mention the route of administration is SC by intra abscess injection.

This has now been added. Lines 299-301 read: “Furthermore, the peptide formulations displayed no aggregation at concentrations as high as 7.5 mg/kg in an in vivo mouse abscess infection model, where they were administered by subcutaneous injection.”

  1. there are occasional non-italicised use of species names eg. line 397, 398 etc. please ensure consistency throughout

We have ensured that species names are consistently italicised.

  1. I noticed that when referencing authors specifically, the reference is given as a date,whereas others are given specific numbers. this seems a little unusual

We have removed the dates to make it consistent throughout the manuscript.

  1. lines 479-485: DOI 10.3390/antibiotics9040155 might be another useful reference here

This has been added. Lines 562-565 read: “Nevertheless, the large number of human monoclonal antibody (mAb) therapies currently being developed and tested in clinical trials to treat bacterial infection, either by direct pathogen targeting or by toxin neutralization [90], highlights the strong potential of these molecules for AMP delivery.”

  1. line 511: use of L- and D- notation should be small CAPS

We have corrected this throughout the manuscript.

5

  1. line 522-536 - seems to be repetition with references (eg. ref 139 and 142) which could be consolidated

This has been done.

Reviewer 3 Report

The review by Suzana K. Straus et al. gives a panoramic vision of how antimicrobial peptides can be modified or encapsulated to obtain improved drugs against bacteria.

The review is comprehensive and up-to-date. I have a few minor comments:

-Line 76, page 2, I would soften the statement, AMP can be modified differently to achieve stability and longer half-life, as discussed further on in the paper.

- “Endogenous targeting”, I would change the title into a less generic one.

-Line 121, page 4, a comparison in MIC values of the vancomycin versus vancomycin nanoparticles would be important.

-Line 193, page 5, It would be important to clarify which specific type of infections the system (photoactivatable daptomycin-loaded polydopamine-coated gold nanocages) could address in the clinical practice, in further developed.

-Line 281, page 7, solid lipid nanoparticles (SLNs) are only cited, either extend the description or remove it.

-Line 319-325, page 8, to give a thorough description of DNA-AMP compound, please better address also the role of DNA.

-3.3. Synthetic Polymers. The authors do not mention oligodendrimeric peptides, such as tetrabranched AMP, that show a very long half-life and interesting antimicrobial and anti-inflammatory activity in vitro and in vivo.

-Check the font of bacteria’s names that has to be always italics.

-Line 441, page 10 “certain AMPs can be 441 manipulated to form dendrimers that display strong antimicrobial activity”, please explain what kind of manipulation.

Author Response

The review by Suzana K. Straus et al. gives a panoramic vision of how antimicrobial peptides can be modified or encapsulated to obtain improved drugs against bacteria. The review is comprehensive and up-to-date. I have a few minor comments: We thank the reviewer for their positive comments.

-Line 76, page 2, I would soften the statement, AMP can be modified differently to achieve stability and longer half-life, as discussed further on in the paper.

We agree and have soften this statement. This now reads, on lines 89-90: “Though few examples of targeted AMP release have been published in literature to date, such release mechanisms can be utilized to improve the specificity, stability and activity of AMP therapeutics, as discussed below.”

- “Endogenous targeting”, I would change the title into a less generic one.

We agree that this may be a generic title, however, this is the term commonly used in literature and is what is covered in this section. If the reviewer have more specific suggestions, we would be happy to consider them.

- Line 121, page 4, a comparison in MIC values of the vancomycin versus vancomycin nanoparticles would be important.

This has now been incorporated and can be found in lines 138-140: “The resulting minimum inhibitory concentrations (MICs) of free vancomycin and the pH-responsive vancomycin nanoparticles were 2.4 and 8.6 μg/mL at pH 6.0, denoting 2.0- and 1.3-fold increases from pH 7.4, respectively.”

- Line 193, page 5, It would be important to clarify which specific type of infections the system (photoactivatable daptomycin-loaded polydopamine-coated gold nanocages) could address in the clinical practice, in further developed.

The clarification has been added. Lines 212-217 now read: “These nanocages released daptomycin in response to near-infrared light irradiation, which itself generated localized bactericidal photothermal effects, thereby combining the localized release of antibiotic with photothermal therapy to eradicate both planktonic and biofilm-associated S. aureus. The authors suggested further development could allow for potential applications in the treatment of orthopedic infections (e.g., surgical debridement), where surgeons would have direct access to 6 the site to facilitate laser irradiation.”

- Line 281, page 7, solid lipid nanoparticles (SLNs) are only cited, either extend the description or remove it.

We have extended the description. Lines 327-339 read: “For example, LL37 was

encapsulated in a NLC for topical wound treatment, where it maintained its immunomodulatory and antimicrobial activity while improving wound healing compared to the free peptide [104].

This is mirrored by work performed by Fumakia and Ho, who found that SLN formulations containing LL37 and the elastase inhibitor Serpin A1 accelerated wound healing and displayed synergistic antimicrobial and anti-inflammatory activities [105]. Sans-Serramitjana et al. also found that encapsulating colistin with NLC maintained the peptide’s antimicrobial activity, which was more stable over time compared to SLN-encapsulated colistin [106].”

- Line 319-325, page 8, to give a thorough description of DNA-AMP compound, please better address also the role of DNA.

We have added the following sentence on lines 381-383: “The authors suggested the aptamers imparted increased cellular uptake of the conjugates, providing improved delivery for targeting intracellular pathogens.”

- 3.3. Synthetic Polymers. The authors do not mention oligodendrimeric peptides, such as tetrabranched AMP, that show a very long half-life and interesting antimicrobial and antiinflammatory activity in vitro and in vivo.

This is presented more extensively now, in section 3.4.2 (which is now separate from the antibodies section in 3.4.3). The first paragraph in this section now reads: “Cell penetrating peptides (CPPs) themselves have been harnessed to improve the antimicrobial activity of AMPs.

Interestingly, these CPPs often negligibly affect the cytotoxicity of the AMPs while improving their antimicrobial activity and specificity, particularly when targeting Gram-negative bacteria [140,141]. Hydrophilic polypeptides (e.g., XTEN and PAS) have also gained much interest in recent years as alternatives to polymers for protein conjugation [26], though no examples of AMP conjugation have yet been published. Recently, short polycationic hexa-arginine polypeptides were conjugated to vancomycin to bypass multiple modes of vancomycin resistance, accompanied by a 1000-fold increase in antibacterial activity [142]. The peptide

conjugates demonstrated efficacy in vivo along with improved biodistribution and excretion profiles compared to free vancomycin. Interestingly, the conjugates displayed no hemolytic activity or cell culture toxicity. Additionally, it has been shown that certain AMPs can be structurally manipulated to form dendrimers by introducing branching peptide chains via reactive amino acid side chain groups (e.g., lysine amines) [88]. These dendritic structures have displayed

strong antimicrobial activity alongside improved serum stability and reduced hemolysis [88].

Likewise, tetrabranched AMP oligodendrimers, also termed multiple antigen peptides, have exhibited strong in vitro and in vivo activity as well as stability against blood proteases. For example, the branched AMP M33-L displayed strong activity against a wide variety of bacteria, and also exhibited LPS neutralization, serum stability and efficacy in mouse sepsis models

[143,144]. Extensive in vivo studies of M33-L also showed the peptide to be much less toxic than the clinically used peptide colistin [144]. Isomerization of the oligodendrimer further 7 improved the AMP’s activity against Gram-positive bacteria, potentially due to increased resistance to bacterial proteases such as staphylococcal aureolysin and elastase [145].”

- Check the font of bacteria’s names that has to be always italics.

This was addressed, also in response to Reviewer 2 (see above).

- Line 441, page 10 “certain AMPs can be 441 manipulated to form dendrimers that display strong antimicrobial activity”, please explain what kind of manipulation.

This has been included now – see the point above.

Reviewer 4 Report

AMPs are a promising solution to the bacterial antibiotic resistance but their therapeutic use is limited. In this paper, Drayton and co, reported an updated and comprehensive review of delivery vehicles used to mitigate the systemic toxicity, the low availability and the rapid degradation of AMPs. The review, describes the status of research and provide important information for other researchers working in this field. The work is well done and should be published on Molecules in its present form.

Typos:

Line 994:  remove reference 149

Author Response

AMPs are a promising solution to the bacterial antibiotic resistance but their therapeutic use is limited. In this paper, Drayton and co, reported an updated and comprehensive review of delivery vehicles used to mitigate the systemic toxicity, the low availability and the rapid degradation of AMPs. The review, describes the status of research and provide important information for other researchers working in this field. The work is well done

and should be published on Molecules in its present form.

We thank Reviewer 4 for their positive comments.

Typos: Line 994: remove reference 149

This has been done.

Round 2

Reviewer 2 Report

The authors are thanked for their revisions, which I think now contribute to a more rounded article. I have nothing further to add, and would be happy to see the article accepted.